# Carrying out embedded implementation research in humanitarian settings: A qualitative study in Cox's Bazar, Bangladesh

**A. S. M. Shahabuddin**[1]*, **Alyssa B. Sharkey**[1], **Debra Jackson**[1], **Paul Rutter**[2], **Andreas Hasman**[2], **Malabika Sarker**[3]

**1** Implementation Research and Delivery Science Unit, Health Section, UNICEF, New York, New York, United States of America, **2** Health Section, UNICEF Regional Office for South Asia, Kathmandu, Nepal, **3** James P Grant School of Public Health, BRAC University, Dhaka, Bangladesh

* ashahabuddin@unicef.org

**Data Availability Statement:** UNICEF Bangladesh owns the data of this research so UNICEF's data sharing policy will be applied to share the relevant

## Abstract

### Background

Embedded implementation research (IR) promotes evidence-informed policy and practices by involving decision-makers and program implementers in research activities that focus on understanding and solving existing implementation challenges. Although embedded IR has been conducted in multiple settings by different organizations, there are limited experiences of embedded IR in humanitarian settings. This study highlights some of the key challenges of conducting embedded IR in a humanitarian setting based on our experience with the Rohingya refugee population in Cox's Bazar, Bangladesh.

### Methods and findings

We collected qualitative data in between January and July 2019. First, we visited Rohingya refugee camps and interviewed representatives from different humanitarian organizations. Second, we conducted interviews with researchers from BRAC University who were engaged with data collection and analysis in a broader embedded IR study on maternal, newborn, child, and adolescent health (MNCAH) program implementation challenges. Data were analyzed using a thematic analysis approach. Two researchers developed and agreed on codes and relevant themes based on the objectives of this study. The findings of this study highlight several challenges encountered while conducting embedded IR in the Rohingya emergency setting in Cox's Bazar, which may have implications for other humanitarian settings. The overall context of the camps was complex, with more than 100 organizations devoted to providing health services for approximately 1 million refugees. Despite the presence of the Bangladesh government, United Nations agencies and other international organizations played key roles in making programmatic and policy decisions for the Rohingya. Because health service delivery modalities and policies and related implementation challenges for MNCAH programs for the refugees changed rapidly, the embedded IR approach used was flexible and able to adapt to changes identified, with research questions and methods modified accordingly. Access to the camps, reaching Rohingya respondents,

data. For accessing study data, please contact: Minjoon Kim (mkim@unicef.org) or Avijit Saha (a. saha@bracu.ac.bd).

**Funding:** This research was funded by Evaluation Section of the UNICEF's Regional Office for South Asia (ROSA), Kathmandu, Nepal. Being an Implementation Research study, staff from UNICEF's Implementation Research and Delivery Science (IRDS) unit in New York Headquarters were involved in study design, supervision and analysis of the data. Evaluation Section from ROSA had no role in study design, data collection and analysis, decision to publish, or preparation of the manuscript.

**Competing interests:** The authors have declared that no competing interests exist.

**Abbreviations:** INGO, international non-governmental organization; IR, implementation research; MNCAH, maternal, newborn, child, and adolescent health; NGO, non-governmental organization.

overcoming language barriers in order to get quality information, and the limited availability of local research collaborators were additional challenges. Working with researchers or research institutes that are familiar with the context and have experience in conducting implementation and health systems research can help with collection of quality data, identifying key stakeholders and bringing them on board to ensure the execution of the project, and ensuring utilization of the research findings. Study limitations include possible constraints in generalizing our conclusions to other humanitarian settings. Implementation research conducted in additional humanitarian settings can contribute to the evidence on this topic.

## Conclusions

Findings indicate that embedded IR can be done effectively in humanitarian settings if the challenges are anticipated, and appropriate strategies and in-country partners put in place to address or mitigate them, before commencing the funding or starting of the project. Understanding the context and analyzing the role of relevant stakeholders prior to conducting the research, considering a simple descriptive method appropriate to answering real-time IR questions, and working with local researchers or research institutes with specific skill sets and prior experience conducting research in humanization contexts may reduce costs and time spent, and ensure collection of quality data relevant for policy and practice.

## Author summary

### Why was this study done?

- Despite a growing body of evidence on the value of research in emergency settings, there is still a knowledge gap regarding the challenges of conducting embedded implementation research (IR) in these settings.

- This study aimed to explore challenges and key lessons learned relating to the design, implementation, and management of embedded IR in humanitarian contexts based on experience with Rohingya refugees in Cox's Bazar, Bangladesh.

### What did the researchers do and find?

- We collected qualitative data from stakeholders supporting and implementing health programs in Cox's Bazar. We also conducted interviews with researchers conducting a broader embedded IR study in Cox's Bazar to obtain their perspectives on challenges in conducting embedded IR in this emergency context.

- The context of the Rohingya refugee camps was complex, with multiple organizations providing maternal, newborn, child, and adolescent health services and playing key roles in formulating programmatic and policy decisions for the Rohingya refugees.

- Health service delivery modalities and policies and related implementation challenges in the camps have changed rapidly since the start of the crisis.

- Access to the camps, reaching Rohingya respondents, overcoming language barriers in order to get quality information, and the limited availability of local research collaborators made conducting embedded IR in Cox's Bazar challenging.

### What do these findings mean?

- This study may help other researchers understand the potential challenges and identify appropriate strategies for conducting embedded IR in humanitarian settings.

- IR studies with simple descriptive methods that are nonlinear and iterative in nature may be most appropriate to answer real-time research questions and, thus, to tackle real-time implementation challenges in humanitarian contexts.

- When planning for embedded IR in a humanitarian setting, it is critical to understand the local context and the role of relevant stakeholders, and to identify key decision-makers to involve in the project in order to ensure access to camps and research participants, and to ensure uptake of research findings and recommendations.

- Working with local researchers or research institutes with local language capabilities and prior experience conducting research in the context may reduce costs and time spent, and ensure collection of better-quality data.

## Introduction

Embedded implementation research (IR) is an approach that can enhance the effectiveness of implementation and scale up of a program by promoting evidence-informed policy and practice [1–4]. Embedded IR bridges the gap between research and policy by embedding relevant decision-makers and program implementers (program managers, frontline health workers, etc.) in the process of research and dealing with issues related to health systems that have relevance in programs and policies [1–6]. This approach focuses on the questions identified by decision-makers and program implementers and ensures that knowledge generated is relevant. As an emerging area of research, embedded IR has been put forward and applied by several countries and organizations as a cornerstone to health systems strengthening and achieving universal health coverage.

While individuals of the ethnic Rohingya minority population in Myanmar have been coming to Bangladesh since 1978 [7–9], a major exodus happened in August 2017. As of January 2019, over 900,000 Rohingya refugees resided in Ukhiya Upazila and Teknaf Upazila (sub-districts of Cox's Bazar District), while the largest single site, the Kutupalong–Balukhali expansion site, hosted more than 600,000 Rohingya [7–12].

Despite substantial progress during past 2 years, Rohingya refugees in Bangladesh remain in a precarious situation. Poor living conditions; lack of safe water, sanitation, and hygiene; high rates of child marriage; lack of support for education; poor healthcare; and lack of livelihoods are major concerns for them [13]. More than 55% of the Rohingya refugees are women and children, many of whom have faced gender-based violence, abuse, trafficking, malnutrition, and serious health problems [14–16]. Utilization of skilled maternal, newborn, child, and adolescent health (MNCAH) services is very low among the Rohingya refugees. One report found that about 78% of Rohingya babies are born in unsafe and unhygienic makeshift shelters

in the district [15]. Coverage of immunization and services to prevent and treat other child health illnesses are also low in the camps [8,10–12,15,17].

Considering the overall situation, UNICEF in collaboration with BRAC University in Bangladesh, undertook an embedded IR project from 3 January to 31 July 2019 with the aim of strengthening MNCAH programs in Cox's Bazar District. The key objectives of the project were to identify key implementation challenges of MNCAH programs, explore potential solutions, and ensure utilization of those solutions for effective implementation of MNCAH programs in the camps through engagement of decision-makers and program implementers. Part of the findings and methods used for the project were published elsewhere [18]. In addition, one of the objectives of the initiative was to document methodological and operational challenges of conducting embedded IR in humanitarian settings, which the present study focuses on.

Several articles have highlighted the challenges of conducting research in humanitarian settings in the context of an emergency [19–21]. However, despite a growing body of evidence on the value of research in emergency settings, there is still a knowledge gap regarding the challenges of conducting embedded IR in these settings [6,21–25]. In recognition of this gap, this present study explored the challenges and key lessons learned relating to the design, implementation, and management of embedded IR in humanitarian contexts based on the Cox's Bazar experience.

## Methods

This study was part of a larger embedded IR project conducted by BRAC University and supported UNICEF. Results of that embedded IR are reported elsewhere [18].

For the current study, a qualitative method was applied to capture the researchers' experience (i.e., challenges) of conducting embedded IR in the Rohingya emergency context in Cox's Bazar. We collected qualitative data in 2 phases to document researchers' experience of conducting IR and to describe potential implications for embedded IR work in other emergency settings.

### Study population and sampling

In January 2019 (first phase), we, the research team (ASMS, AH, and MS), visited a UNICEF-supported primary healthcare center and a health post within the Kutupalong refugee camps in Ukhiya Upazila and interviewed representatives from non-governmental organizations (NGOs), United Nations (UN) organizations, and the Bangladesh government who have been supporting and implementing MNCAH programs in Cox's Bazar. Questions were asked to document the geographic location of the camps, living conditions of the Rohingya refugees, MNCAH service delivery modalities, and related policies and roles and responsibilities of different stakeholders.

To complement the data of the first phase, in the second phase (July 2019, following completion of the larger embedded IR project), we interviewed 5 purposively selected members of the BRAC University research team. We asked about their experiences conducting IR in Cox's Bazar, challenges they encountered while collecting data, and their suggestions for future embedded IR in similar settings. Table 1 shows the data collection methods and types of respondents interviewed.

### Data collection and analysis

Three researchers (ASMS, AH, and MS) conducted all interviews for this study and took notes while observing the Kutupalong Rohingya refugee camps of Ukhiya Upazila in Cox's Bazar.

**Table 1. Data collection methods and types of respondents.**

| Time of data collection | Data collection methods | Respondents (*n*) |
|---|---|---|
| Phase 1 (January 2019) | Observation within Rohingya camps (1 health post and 1 primary healthcare center) | — |
| | In-depth interviews | Representatives from the United Nations High Commissioner for Refugees, International Labour Organization, and United Nations Population Fund (*n* = 3) |
| | | Representatives from national non-governmental organizations (*n* = 2) |
| | | Government official (*n* = 1) |
| | | UNICEF field staff (*n* = 1) |
| Phase 2 (July 2019) | In-depth interviews and focus group discussion | BRAC University research team members (*n* = 5) |

The main objectives for visiting the camps were to document living conditions, available health facilities, accessibility of the health facilities, number of organizations working in the camps and their roles, and service delivery mechanisms of programs designed to meet the needs of the Rohingya. During the camp visits, the research team conducted interviews with different stakeholders supporting and implementing MNCAH programs in Cox's Bazar. We asked questions relating to challenges and facilitators of MNCAH program implementation, existing research gaps, and respondents' views on the need for IR on MNCAH programs in Cox's Bazar. During the second phase, we interviewed BRAC University researchers to understand their experiences conducting research in the camps as well as methodological or operational challenges they experienced while conducting IR and their suggestions for future IR studies in emergency settings.

Interviews with UN representatives were done in English; other interviews were done in Bengali. Data collected from observations and interviews were transcribed in English. ASMS took extensive notes during the visit to the camps and during the interviews with BRAC University researchers. Thematic analysis was used to analyze and present data [26–28]. After reading field notes and transcripts, ASMS developed codes and relevant themes based on the objectives of this study (see S1 COREQ Checklist). To increase validity, AH also read a sample of transcripts, and consensus was reached after discussion to finalize themes. The main IR protocol obtained ethical approval from the Ethical Review Committee of BRAC University. Before conducting each interview, we obtained verbal informed consent from all participants. In addition, written permission was obtained from the Office of the Refugee Relief and Repatriation Commissioner (RRRC) prior to accessing the camps and conducting observations.

## Results

Five key themes emerged from the observation and interview data: (1) unique and complex context with multiple actors, (2) adaptation due to a dynamic situation and changing service delivery modalities, (3) difficulties accessing the camps and research participants, (4) language and other barriers to accessing quality information, and (5) need for experienced in-country research partners.

### Unique and complex context with presence of multiple actors

Geographic location, the high number and density of the population, the severity of the emergency, the broad range of activities underway by various organizations, the lack of

coordination among different actors, and an absence of a long-term plan for the population makes the situation in Cox's Bazar unique and complex. During the January 2019 field visit, researchers documented that more than 100 organizations (including local organizations, national organizations, international NGOs [INGOs], donor agencies, and UN organizations) were working to serve nearly 1 million Rohingya refugees with shelter, food, education, health, and other emergency services. Respondents mentioned that despite the presence of the Bangladesh government, UN agencies and several INGOs were playing key roles in making programmatic and policy decisions for Rohingyas. One government official and 2 representatives from national NGOs mentioned the lack of coordination and clarity in terms of the roles and responsibilities of different organizations.

> "There is a coordination committee at the upazila [sub-district] level. Representatives from all the organizations working in Cox's Bazar are supposed to participate in the monthly meeting of that committee but many of the organizations do not participate."—government official

Because of the strong presence and authoritative role that various stakeholders had in determining local programs and policies, a steering committee was formed to guide the embedded IR, which was composed of representatives of government and other key organizations (UN agencies and INGOs). However, it was difficult to have regular meetings with all members due to their heavy workloads for the emergency response.

> "Organizing regular meeting with all stakeholders is difficult here. All of us are busy. I want to be part of the embedded research, but I have so many other things to do."—representative of an NGO

Further, while IR often aims to find sustainable solutions to implementation challenges—particularly solutions that can strengthen the existing health system—interviews and meetings with stakeholders revealed the difficulty of achieving this aim with IR in the Rohingya context, where the Bangladesh government has not made a commitment to ensuring long-term services for the refugee population.

## Adaptation due to frequent changes in service delivery modalities, existing policies, and the overall situation in the camps

Respondents mentioned that since the major influx of Rohingya refugees, which occurred in August 2017, there have been continuous changes in service delivery modalities and policies to meet the refugees' current needs. Initially, given the severity of the crisis, different organizations struggled to meet the rapidly rising demand for basic services. However, several respondents mentioned that this demand has grown more acute over time.

> "Situation here is changing every day including the challenges. The problem we have today may change tomorrow."—representative of a UN organization

Respondents also mentioned that since the beginning of the crisis, the characteristics of the emergency response have evolved from a response with heavy involvement from communities, NGOs, INGOs and government departments to one with reduced presence of national NGOs and INGOs over time.

One key suggestion of several respondents was that the IR should adapt to address the ongoing implementation challenges of MNCAH programs. For example, since the outset of

the emergency, the structural, financial, human resources, social, and political situations of the camps have been changing rapidly. Therefore, the approach of IR should be flexible, with research questions and methods that can be modified as needed, as in a grounded theory approach. Respondents also suggested that it is critical that approaches and methods used enable quick results, to help fill immediate knowledge gaps of the MNCAH programs as they arise.

## Difficulty accessing the camps and Rohingya research participants

During observations, we noted that most of the camps are built in hilly areas that are highly vulnerable to floods, landslides, and cyclones. All members of the research team noted that access to the community and different health facilities, and finding relevant respondents, was a major challenge, particularly during the rainy season. Due to heavy rainfall and bad conditions of roads, it was not possible to get into the camps with vehicles. The research team therefore had to walk long distances to reach the camps, at times struggling because of recent landslides. Further, they reported that, due to the bad road conditions inside the camps, the patient flow at health facilities was very low. There was no accommodation facility near to the camps, so the research team had to commute each day from the city of Cox's Bazar to the camps. Due to low patient flow, the research team had to travel to the camps multiple days to find appropriate respondents for interview. A few members of the team mentioned that the timing of data collection was not good. They suggested collecting data outside of the rainy season in order to save time and costs and to reduce health risks to the research team.

> "It was difficult to go into the camps due to bad road condition. It has been raining for the past few days which made the situation even worse. We should have thought about it earlier and collected data some other time."—member of BRAC University research team

## Language and other barriers to accessing quality information

Communication with research participants, particularly those from the Rohingya community, was a challenge mentioned by all the interviewed members of the BRAC University research team. For the overall IR project, the research team had to interview several community health workers, supervisors of community health workers, and Majhis (Rohingya community leaders) who were Rohingya refugees. Although 2 local research assistants who understood the Rohingya language were recruited to interview and collect data from Rohingyas, they reported having difficulties getting information from Rohingya participants. In particular, the research team mentioned 2 issues: (1) it was difficult to find research assistants who were proficient in speaking and understanding the Rohingya language and who were willing to work in Cox's Bazar, and (2) there were challenges to finding skilled research assistants who not only spoke the language but were also experienced in conducting qualitative interviews and focus group discussions. The BRAC University research team mentioned that most of the Rohingya respondents were not comfortable speaking with them.

> "We recruited two research assistants who knew Rohingya language. Still many Rohingya participants were not comfortable to speak with them as they [the research assistants] were not from the local community. Moreover, research assistants were not familiar with conducting qualitative interviews."—member of BRAC University research team

### Need for experienced in-country research partners

Respondents emphasized the importance of collaborating with researchers or research institutes that are familiar with the camp context and that have experience in conducting IR. They highlighted that it would be very difficult for any researchers or institutes to conduct IR in such a context if they are not familiar with the systems, policies, and potential challenges to accessing the camps and respondents. Without adequate experience working in the humanitarian context, researchers may experience several challenges while collecting data, identifying key stakeholders, and bringing them on board to ensure the execution of the project and to ensure utilization of the research findings by key stakeholders.

## Discussion

More than 100 organizations have been involved in providing MNACH services in Cox's Bazar since August 2017. Despite the presence of the Bangladesh government, UN agencies and other INGOs have been playing key roles in formulating programmatic and policy decisions for Rohingya refugees. Over the study period, rapid change has been observed in delivering health services and formulating policies, as well as variation in the implementation challenges of MNCAH programs. Therefore, the approach for embedded IR had to be flexible to adapt to changes identified, with research questions and methods modified accordingly. In addition, access to the camps, reaching Rohingya respondents, overcoming language barriers in order to get quality information, and the limited availability of local research collaborators were highlighted as additional challenges by the respondents.

Every emergency is unique in terms of the severity of the problem, population affected, geographic and political situations, modalities of service provision, and the number and types of stakeholders involved in provision of services. Similarly, the need for, appropriateness of, and use of research in such settings will vary.

Embedded IR is distinct from other types of research in that decision-makers and program implementers are actively engaged in research design, data collection, analysis, and adoption of findings [1,4,29–31]. Active engagement of program implementers at the outset and throughout the research project is critical. A key question is whether it is reasonable to do embedded IR in an emergency setting such as the Rohingya refugee crisis, bearing in mind the additional commitment that active involvement in embedded IR requires of implementers who are already under significant pressure. Both practical and ethical factors must be taken into consideration in any particular emergency. As part of the planning for IR in humanitarian settings, it is therefore crucial to have an explicit understanding of the context, to identify key stakeholders, and to analyze their roles and capacity in programmatic policy decisions and service provision [1,3,4,29,32].

In the context of Cox's Bazar, an initial visit to the camps and interviews with various stakeholders were helpful to map the health situation, geographic locations, and available resources and to assess the roles of different actors. Insights from this initial assessment helped the BRAC University research team to identify and develop appropriate context-specific research tools (interview, observation guidelines, etc.), to recruit local research assistants who could conduct interviews in the language of the target population of Rohingyas, and to know which organizations and stakeholders should be part of the research process to ensure subsequent uptake of research findings and recommendations.

The embedded IR approach that UNICEF and other organizations have used typically engages decision-makers and implementers, particularly officials from the Ministry of Health and program implementers, in the research process [33]. However, the initial assessment of implementers' capacity and their availability to participate in the research suggested that the

model for research in the emergency setting would have to diverge from the traditional model for embedded IR. In the context of Cox's Bazar, considering the active role of UN organizations and INGOs such as Médecins Sans Frontières and Save the Children, the research teams also engaged those organizations to ensure utilization of the IR findings. Further, it was decided that instead of direct and continuous involvement in design, data collection, analysis, etc., the implementers would be periodically consulted and given opportunities to provide input at critical stages in the research project. This proved to be more appropriate to ensure their active engagement, while also respecting the level of involvement they felt was feasible for them.

As IR deals with real-time implementation problems, so the method chosen to carry out any particular study should fit the purpose and should address the relevant and real-time questions identified. It should provide evidence that can be used for real-time program improvement in a dynamic and nonlinear way [1,4,29,34,35]. Like in other emergency settings, the overall situation of Rohingya refugees has changed rapidly, as have the barriers to implementing MNCAH services [36]. The scope of the response itself has evolved from addressing the urgent needs of refugees, to considering the needs of host communities, the risks of a "crisis within a crisis" posed to both groups during the monsoon season, and medium-term planning [8,9,12,37]. Given such an unstable situation, and to align IR with existing MNCAH programs, a relatively easy and flexible research method that could produce quick, real-time data was pivotal. Therefore, instead of using a complex experimental design, a simple descriptive design with both qualitative and quantitative approaches may be more feasible and appropriate in other such settings [19–21]. In addition, an adaptive and iterative approach involving relevant stakeholders may be more appropriate to understand the situation in each stage of the emergency, learning and adjusting the methodology accordingly, as required [38]. In some cases, analysis of routinely collected secondary MNCAH data may be helpful for answering questions related to implementation barriers, in which case collection of primary data may not be necessary.

Gaining access to certain areas and reaching potential respondents are common research challenges in humanitarian settings. In Cox's Bazar, due to several recent incidents in the camps, the government of Bangladesh put access restrictions on research organizations. Before going to the camps, every research organization needed government approval. Moreover, the overall condition of the camp settlements in Cox's Bazar was very challenging. Almost 70% of locations were accessible only by footpath, which created an extremely challenging situation for the delivery of humanitarian services [36]. Being a hilly and coastal area, the sites were highly vulnerable to rain, floods, cyclones, fire, and landslides, a situation made worse during the rainy season [8,10,36]. In our study, the research team had to visit the sites multiple times to get respondents during the rainy season, which cost additional time and money. Further, the safety of the research teams is an important consideration in humanitarian settings as difficult terrain, weather, and violence are often problems, as has been highlighted in other studies [19,21,39,40].

Further, language is often a challenge to researchers working with linguistically and culturally diverse communities [41]. The language barrier may result in failure to understand the true situation of certain communities or may result in failure to include certain groups in a research project, especially when this barrier proves insurmountable. This barrier can be mediated through the use of an interpreter or translator or bilingual researchers [21,39]. However, it becomes difficult to collect high-quality information when a bilingual researcher does not have sufficient research skills. In our study, the BRAC University research team had to spend substantial time on training to build the capacity of the local bilingual research assistants to make them familiar with the interview guidelines. Moreover, during the interviews, bilingual research assistants had to be supported by senior researchers. These factors increased the amount of both time and money needed to complete the research.

Challenges of conducing IR in humanitarian settings can be reduced with the involvement of researchers or research institutions with adequate research capacity and understanding of the context and research approach. Although the BRAC University research team was familiar with the context and with the IR approach, they still had challenges collecting data due to difficulties accessing the camps, finding respondents on time, and finding good bilingual researchers to do interviews with the Rohingya participants. In general, research capacity within humanitarian contexts can be a challenge, particularly in low-income countries that lack researchers or research institutes with the required research skills and knowledge of local settings [42,43]. This research therefore often depends on international researchers or research organizations despite the fact that this can require more funding and time than working with a local research institution. Further, IR is still a relatively new area for many researchers in low-income settings. However, investing in collaborations with in-country researchers or research institutes provides opportunities to build their capacity for IR in humanitarian contexts, which may help to institutionalize IR in emergency settings. This type of collaboration may also reduce research costs and time and increase the likelihood of obtaining reliable data that are useful for policy and programmatic decisions in the local context.

## Study limitations

The findings presented in this study are based on the experience of conducting embedded IR in the Cox's Bazar Rohingya emergency setting. Although this study highlights several key considerations for conducting embedded IR in humanitarian settings, the findings may not be applicable to other humanitarian settings because of the uniqueness of every emergency. Sharing of experiences and lessons learned from IR approaches used in other emergency settings will be useful to enhance our collective understanding of what is appropriate and helpful to program implementers.

## Conclusions and recommendations

Although research is important to improve the health of a population affected by an emergency, researchers often face challenges in conducting research in such settings. Understanding these challenges can enable researchers to adopt appropriate strategies before commencing the study. This study identified 5 major challenges, each of which can be addressed or at least mitigated if it is anticipated in advance. First, understanding the context and analyzing the role of relevant stakeholders are prerequisites to mapping potential operational challenges and to identifying key decision-makers to involve in the project to ensure uptake of research findings and recommendations. Second, instead of using complex experimental designs, IR with simple descriptive methods that are nonlinear and iterative in nature may be more appropriate to answer real-time research questions and, thus, to tackle real-time implementation challenges. Third, planning needs to pay close attention to how access to refugee camps and research participants will be achieved. Fourth, recruitment of researchers who can speak the language of, and are acceptable to, the study population is key. And, fifth, working with local researchers or research institutes with specific skill sets and prior experience conducting research in the humanitarian context may reduce costs and time spent, and may ensure collection of better-quality data that are relevant for local policy and practice.

## Supporting information

**S1 COREQ Checklist.**
(DOCX)

## Acknowledgments

The authors would like to acknowledge colleagues from UNICEF's Cox's Bazar office (Yulia Widiati, Abdullah Bin Akhtar, ASM Mainul Hasan, and Helen Chakma) for their support in the field during data collection, and colleagues at the UNICEF Bangladesh Country Office (Maya Vandenent and Juanita Vasquez Escallon) and ROSA (Samuel Bickel) for their useful feedback on the proposal. The authors would also like to thank the study participants and participating agencies in Cox's Bazar.

## Author Contributions

**Conceptualization:** A. S. M. Shahabuddin, Alyssa B. Sharkey, Debra Jackson, Andreas Hasman, Malabika Sarker.

**Data curation:** A. S. M. Shahabuddin, Alyssa B. Sharkey, Debra Jackson, Paul Rutter, Andreas Hasman, Malabika Sarker.

**Formal analysis:** A. S. M. Shahabuddin, Andreas Hasman, Malabika Sarker.

**Funding acquisition:** Malabika Sarker.

**Investigation:** A. S. M. Shahabuddin, Malabika Sarker.

**Methodology:** A. S. M. Shahabuddin, Alyssa B. Sharkey, Debra Jackson, Paul Rutter, Andreas Hasman, Malabika Sarker.

**Project administration:** Malabika Sarker.

**Resources:** Malabika Sarker.

**Software:** A. S. M. Shahabuddin.

**Supervision:** A. S. M. Shahabuddin, Alyssa B. Sharkey, Debra Jackson, Malabika Sarker.

**Validation:** A. S. M. Shahabuddin, Debra Jackson, Malabika Sarker.

**Visualization:** A. S. M. Shahabuddin.

**Writing – original draft:** A. S. M. Shahabuddin.

**Writing – review & editing:** Alyssa B. Sharkey, Debra Jackson, Paul Rutter, Andreas Hasman, Malabika Sarker.

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
