## [Decision Letter · Decision Letter 0]

4 Feb 2020

Dear Dr. Shahabuddin,

Thank you very much for submitting your manuscript "Embedded implementation research in humanitarian settings: Learnings from an experience with the Rohingya emergency in Cox’s Bazar, Bangladesh" (PMEDICINE-D-19-03644) for consideration at PLOS Medicine for our upcoming special issue on refugee and migrant health. 

Your paper was evaluated by the editors and sent to independent reviewers, including a statistical reviewer. The reviews are appended at the bottom of this email and any accompanying reviewer attachments can be seen via the link below:

[LINK]

In light of these reviews, we will not be able to accept the manuscript for publication in the journal in its current form, but we would like to invite you to submit a revised version that fully addresses the reviewers' and editors' comments. You will appreciate that we cannot make a decision about publication until we have seen the revised manuscript and your response, and we expect to seek re-review by one or more of the reviewers. 

We hope to receive your revised manuscript by Feb 25 2020 11:59PM. Please email us (plosmedicine@plos.org) if you have any questions or concerns.

Please let me know if you have any questions. Otherwise, we look forward to receiving your revised manuscript soon. 

Sincerely,

Richard Turner, PhD

rturner@plos.org

In the data statement, you note that "some restrictions [on data availability] will apply". Please explain what these restrictions are (we can imagine that reporting interview transcripts in a humanitarian setting might be potentially problematic, for example). 

Please restructure your title to match journal style. We suggest "Carrying out embedded implementation research in humanitarian settings: a qualitative study in Cox's Bazar, Bangladesh". 

Please combine the "methods" and "findings" subsections of your abstract. The final sentence of the new combined subsection should summarize the study's main limitations. 

Please begin the sentence at line 61 with "Our findings indicate that ..." or similar. 

After the abstract, we will need to ask you to add a new and accessible "author summary" section in non-identical prose. You may find it helpful to consult one or two recent research papers published in PLOS Medicine to get a sense of the preferred style. 

Early in the methods section of your main text, please state whether the study had a protocol or prespecified analysis plan, and if so attach the relevant document(s) as a supplementary file (referred to in the text). Please highlight analyses that were not prespecified. 

Please restructure the early part of the discussion section of your main text, as the first paragraph should provide a summary of the paper's findings. 

In the discussion section, please add a discrete paragraph on study limitations. 

Throughout your text, please style reference call-outs as follows: "... an emergency [18-20]."

In your reference list, please ensure that journal names are abbreviated consistently (e.g., "Lancet" for reference 1).

Please add a completed checklist for the most appropriate reporting guideline, which may be COREQ, as a supplementary file, referred to in your methods section. In the checklist, please refer to individual items by section (e.g., "Methods") and paragraph numbers rather than by page or line numbers, as the latter generally change in the event of publication. 

Comments from the reviewers:

*** Reviewer #1: 

The authors are correct in identifying both the gap and the potential for research in a humanitarian emergency. They are also correct in identifying the unique situation in which the Rohingya in Bangladesh find themselves, sheltered in a set of camps that form the largest migrant population in the world. Although they briefly explain the fundamentals of embedded implementation research and proceed to establish the difficulties of conducting such research in a crisis situation, it would seem prudent to outline the process for conducting a successful IR project at the outset, and then later compare it to their own observations. 

Although the magnitude of the current crisis is alluded to (in terms of migrant numbers), its scale can be further quantified by mentioning the total number of camps present as well as the total number of primary health-care facilities in them. This data may help the authors further explain some of the difficulties they faced in conducting this project. However, this will highlight the very small sample size they have taken in Phase 1. 

The culmination of the study seems to rest on the five challenges identified; as it stands, it could be argued that these challenges could have been identified by analysing secondary data at a much more lower cost. Where this study would find its strength is in providing specific solutions for these challenges. A prime example would be in recognising the potential of the doctors in the primary care facilities as both as a translator and as researchers in the field.

*** Reviewer #2: 

[See attachment]

Michael Dewey

*** Reviewer #3: 

Thank you for the opportunity to review the manuscript entitled, "Embedded implementation research in humanitarian settings: Learnings from an experience with the Rohingya emergency in Cox's Bazar, Bangladesh." The article has the potential to provide an important contribution as it focuses on the intersection of implementation research and humanitarian contexts, where there is a dearth of existing research. 

My main comments focus on the framing, methods and results, and I think these points require considerable attention to maximize the potential of the paper. 

Framing: 

- It seems that this is a study OF embedded implementation research, and that the methods used for the study OF embedded implementation research were not necessarily "embedded." It is important to assert more clearly how this study fits with the broader IR study, as it is not clear from the text. It is clear that the authors conducted interviews and observations, but not clear if that is separate and in addition to the broader IR study, with the clear intention of investigating the IR process, or something else. It also does not seem that the present study is itself an embedded IR, and if so the purpose and choice of methods should be clearly stated.

- Since IR--and particularly embedded IR--involves high levels of interaction, co-design, etc. with the implementors, it is important to clearly identify who the implementers are, and how they appear in the study. (presumably the "we" is the authors, but what role do the authors play in actual service delivery? and if people involved in direct service delivery are not authors, what role did they play?) It is reasonable to exclude the names of specific organizations or actors from the manuscript, but it is not clear who from the research team is an implementer, since the UNICEF-supported health posts are run by implementing partners, which could include BRAC, but it is unclear what role BRAC University has, or what relationship they have to the implementing partners. 

- Since this manuscript is distinct from the broader IR project and results, I think it needs to be clear what is the specific contribution of this study. If it is focused on the complexity of the Rohingya crisis, that is important and valid. However, the article seems to oscillate in its claims about what it is aiming to achieve. If it is about conducting embedded IR within the Rohingya crisis, then it seems that it should focused on the specific barriers/facilitators that enable rigorous IR. 

Methods:

- the methods section requires strengthening. Why were qualitative methods chosen for this study? Why Thematic Analysis? What epistemological approach was used in the design and analysis?

Results: 

- As alluded to above, all of the findings are valid and important observations of the complexity of the Rohingya crisis, but it is not clear how they inform our understanding of the feasibility of embedded IR, or how embedded IR should be implemented. The connection from results such as "language barriers" for how that would affect IR, but the relationships between these findings and IR is not clearly or explicitly stated.

- My understanding of embedded IR is that it is, by definition, a collaborative, adaptive approach (e.g. Churruca et. al, 2019), so it is confusing to read statements such as: "It was therefore recognised that the embedded IR approach had to be flexible and able to adapt to changes identified, with research questions and methods modified accordingly." Such statements should be framed as reinforcing the need for a truly embedded approach.

Thanks again and I hope these comments are helpful.

***

[LINK]

---

## [Decision Letter · Decision Letter 1]

5 May 2020

Dear Dr. Shahabuddin,

Thank you very much for re-submitting your manuscript "Carrying out embedded implementation research in humanitarian settings: a qualitative study in Cox's Bazar, Bangladesh" (PMEDICINE-D-19-03644R1) for consideration at PLOS Medicine.

I have discussed the paper with editorial colleagues and it was also seen again by two reviewers. I am pleased to tell you that, provided the remaining editorial and production issues are dealt with, we expect to be able to accept the paper for publication in the journal.

[LINK]

Please let me know if you have any questions. Otherwise, we look forward to receiving the revised manuscript shortly. 

Kind regards,

Richard Turner, PhD

rturner@plos.org

Requests from Editors:

Currently, your data statements ("No - some restrictions will apply" and "all relevant data are in the manuscript") appear contradictory. If you are unable to provide full underlying data in the ms and supplementary files, please supply non-author contact details for readers interested in inquiring about access to study data (e.g., for a [non-author] data contact person or repository at UNICEF). 

Please revisit your abstract, aiming to improve clarity and readability. We suggest that you revise the later part of the "methods and findings" subsection of the abstract to explicitly list the 5 themes that you focus on in the results section of the main text (around lines 220-225), devoting a sentence, say, to each theme in the abstract.

Please add study dates to the "methods and findings" subsection of your abstract; the final sentence of this subsection should summarize 2-3 of the study's main limitations.

Please trim the "author summary" section so that the three subsections consist of no more than 3-4 points, with the individual points comprising no more than 1-2 short sentences. 

At line 63, please make that "findings indicate".

At line 320, please add a few words to indicate the timeframe of when the services were provided in Cox's Bazar.

Please add some additional quotations to the results section, as requested by referee 2. 

In your reference list, please add full access details to references 19 and 37.

Please remove all iterations of "[Internet]" from the reference list.

You mention a COREQ checklist, but we did not find an attachment with your submission. Please enclose a completed checklist with the forthcoming version of your ms, as a supplementary document entitled "S1_COREQ_Checklist or similar, and referred to in your methods section (i.e., "See S1_COREQ_Checklist"). 

Individual items in the checklist should be referred to by section (e.g., "Methods") and paragraph numbers rather than by page or line numbers, as the latter generally change in the event of publication. 

Comments from Academic Editor:

I do not feel that the authors have adequately described their embedded implementation research methods: precisely how decisionmakers and implementers have impacted implementation challenges.

Comments from Reviewers:

*** Reviewer #1: 

The changes made make for a comprehensive and much smoother read. The reason for this submission is now more apparent and the discussion provides valuable insight in to embedded IR in a new setting. 

*** Reviewer #2: 

The authors have addressed my points. I still feel that more quotations in the text from the respondents would have been helpful but I am not a specialist in text-based studies.

Michael Dewey

***

[LINK]

---

## [Editor Report · Decision Letter 2]

15 Jun 2020

Dear Mr. Shahabuddin, 

On behalf of my colleagues and the academic editor, Dr. Terry McGovern, I am delighted to inform you that your manuscript entitled "Carrying out embedded implementation research in humanitarian settings: a qualitative study in Cox's Bazar, Bangladesh" (PMEDICINE-D-19-03644R2) has been accepted for publication in PLOS Medicine. 

PRODUCTION PROCESS

PRESS

PROFILE INFORMATION

Thank you again for submitting the manuscript to PLOS Medicine. We look forward to publishing it. 

Best wishes, 

Richard Turner, PhD

Senior Editor 

PLOS Medicine

plosmedicine.org